# Nonparametric Regressive Point Processes Based on Conditional Gaussian Processes

**Siqi Liu**
Department of Computer Science
University of Pittsburgh
Pittsburgh, PA 15213
siqiliu@cs.pitt.edu

**Milos Hauskrecht**
Department of Computer Science
University of Pittsburgh
Pittsburgh, PA 15213
milos@pitt.edu

## Abstract

Real-world event sequences consist of complex mixtures of different types of events occurring in time. An event may depend on past events of the same type, as well as, the other types. Point processes define a general class of models for event sequences. "Regressive point processes" refer to point processes that directly model the dependency between an event and any past event, an example of which is a Hawkes process. In this work, we propose and develop a new nonparametric regressive point process model based on Gaussian processes. We show that our model can represent better many commonly observed real-world event sequences and capture the dependencies between events that are difficult to model using existing nonparametric Hawkes process variants. We demonstrate the improved predictive performance of our model against state-of-the-art baselines on multiple synthetic and real-world datasets.

## 1 Introduction

Event sequences consist of timestamps of events occurring over a period of time. They arise commonly in our everyday life. Examples are spread of news in a social network, buying and selling actions in a stock market, occurrences of earthquakes in a region, administration of medications for a patient, and many others. Because of their wide applicability, event sequences and their models have become popular in machine learning research.

Point processes [Daley and Vere-Jones, 2003, 2007] can model event sequences by representing events as points in the one-dimensional space: time. In general, a point process defines a probabilistic distribution of points in a space. For a (temporal) point process, the distribution is uniquely determined by its intensity function, which defines the rate of event occurring at any instant.

Two main types of point process models have been developed independently over years. One type is what we call "regressive point processes", where the dependencies of the intensity function on the past events are *directly* modeled. Hawkes processes [Hawkes, 1971] are the most studied and used class of regressive point processes (e.g., [Zhou et al., 2013a],[Zhou et al., 2013b],[Bacry and Muzy, 2014],[Lee et al., 2016],[Wang et al., 2016],[Xu et al., 2016],[Eichler et al., 2017]). A benefit of the regressive point processes is that they are easy to apply and interpret. They can be learned on a set of sequences and then applied on another *unseen* set of sequences. Since the influence of each past event on the intensity is explicitly modeled, it is easy to see how different types of events influence each other over time.

Another type is what we call "latent-state point processes", where the dependencies of the intensity on the past events are *indirectly* modeled through a latent state. Based on the past latent state, we can infer the future latent state and thereby predict future events. The most studied class of latent-state point processes are Gaussian-process-modulated point processes (e.g., [Adams et al., 2009],[Lasko,

2014],[Rao and Teh, 2011],[Gunter et al., 2014],[Lloyd et al., 2015],[Lloyd et al., 2016],[Ding et al., 2018],[Kim, 2018]). They use some transformation of a Gaussian process (GP) as the prior for the intensity function, which provides a probabilistic distribution over possible intensity functions and acts as the latent state. Then the posterior of the intensity function can be inferred from the data. The main benefit of GP-modulated point processes is that they provide a principled way to flexibly model the intensity functions. However, a significant drawback is that they are harder to apply, compared with Hawkes processes, due to the need of inferring a separate latent state for *each* sequence. To make inference on any new sequence, long enough history of the sequence must be available. It is impossible to learn a model from a set of sequences and apply it to other unseen sequences (i.e. cold start).

In this work, we propose a new nonparametric model, GP regressive point process (GPRPP), combining the advantages of the above two models: the flexibility of GP-modulated point processes and the applicability of Hawkes processes. (A discussion of related work is in the supplementary material.) Similar to Hawkes processes, our model directly captures the dependencies of the intensity function on the past events. However, unlike Hawkes processes, the dependencies are modeled nonparametrically through a GP. Meanwhile, different from GP-modulated point processes, the input of our GP is not defined by the "absolute" time relative to each sequence, but by the collection of "relative" times from past events of different types. This defines a latent state independent of specific sequences, and therefore can be learned from and applied to different sequences. Figure 1 illustrates the differences between GPRPP and the previous models.

To better model the dependencies of the intensity function on the past events, we propose a conditional GP model for GPRPP. It relies on a set of points introduced in the input space of the GP to capture the dependencies independent of specific sequences. These points, although bearing a similarity to the inducing points in sparse GPs [Quiñonero-Candela and Rasmussen, 2005, Titsias, 2009], function quite differently, because instead of marginalizing them out, we condition on them for inference.

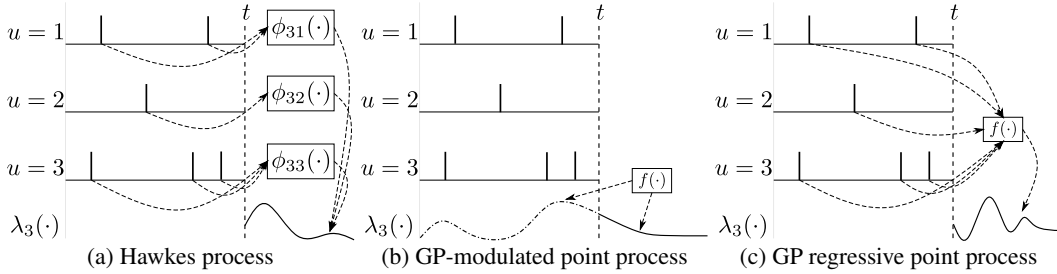

(a) Hawkes process      (b) GP-modulated point process      (c) GP regressive point process

Figure 1: Illustrations of different point process models. The first three rows are a multivariate event sequence consisting of events (stems) of three types ($u$) on the timeline. The vertical line $t$ marks the current time. The last row is the estimated conditional intensity function (CIF) $\lambda_3(\cdot)$ for event type $u = 3$. (a) In a Hawkes process, the CIF depends on the past events through the triggering kernels $\phi_{u_i u_j}$ (Eq. 1). (b) In a GP-modulated point process, the entire (transformed) CIF on the specific sequence is a function with a GP prior. (c) In a GPRPP, the (transformed) CIF depends on the past events through a function with a GP prior.

## 2 Preliminary: point processes

The training data consist of multiple sequences $\mathcal{D} = \{y_c\}_{c=1}^{|\mathcal{D}|}$. Each $y_c$ is a sequence of (time, label) pairs $y_c = \{(t_i, u_i)\}_{i=1}^{|y_c|}$, representing the time and the type of each event, where $t_i \in \mathbb{R}_{\geq 0}$ and $u_i = 1, \ldots, U$. A (temporal) point process has a conditional intensity function (CIF) $\lambda(t) = \lim_{dt \to 0^+} \frac{\mathbb{E}[N([t, t+dt))|\mathcal{H}_t]}{dt}$ as the instantaneous rate of events at time $t$ given the history $\mathcal{H}_t$ up to $t$, where $N(\cdot)$ counts the number of events in an interval. For example, for a Hawkes process, the CIF of event type $u_i$ is defined as

$$\lambda_{u_i}(t) = \mu_{u_i} + \sum_{t_j < t} \phi_{u_i u_j}(t - t_j) \tag{1}$$

where $\phi_{u_i u_j}$ is a function of time that characterizes the influence of past events of type $u_j$ on type $u_i$. It is called a *triggering kernel* in previous works. Meanwhile, $\mu_{u_i}$ defines the baseline intensity.

Given the CIF, the probability density of the data $\mathcal{D}$ is

$$p(\mathcal{D}) = \prod_{c=1}^{|\mathcal{D}|} p(y_c) = \prod_{c=1}^{|\mathcal{D}|} \prod_{i=1}^{|y_c|} \lambda_{u_i}(t_i) \exp\left(-\int_{t_0^c}^{T^c} \sum_{u=1}^{U} \lambda_u(t) dt\right) \tag{2}$$

where $t_0^c$ and $T^c$ are the start and end time of $y_c$. Without loss of generality, we assume $t_0^c = 0$ from now on. We note that the density of the data factorizes over the individual sequences $y_c$ (Eq. 2), while the density of each sequence factorizes over the event types (Eq. 3):

$$p(y_c) = \prod_{u=1}^{U} \left[\prod_{i=1}^{|y_c|} \lambda_u(t_i)^{\delta(u_i, u)} \exp\left(-\int_{t_0^c}^{T^c} \lambda_u(t) dt\right)\right] \triangleq \prod_{u=1}^{U} p_u(y_c) \tag{3}$$

where $\delta(x, y) = 1$ when $x = y$, and $0$ otherwise. In this way, a point process can be viewed as a set of sub-models, one for each type of events. The total likelihood is the product of the likelihood of all the sub-models. For a Hawkes process, each sub-model is similar to a regression model, where the predictors are the elapsed times since the past events of all types, transformed through the triggering kernels.

## 3   GP regressive point processes

Inspired by the regressive view of Hawkes processes, we propose a new model based on Gaussian processes (GPs). Since the density factorizes over sequences (Eq. 2), we describe the method for one sequence $y_c$ in the following. To avoid cluttering, we use $y$ to denote the sequence. Since the density of each sequence factorizes over event types (Eq. 3), we describe the method for $p_{\tilde{u}}(y)$ of one type $\tilde{u}$ (*target type*) and call the events of type $\tilde{u}$ the *target events*. The same method can be repetitively applied to all $p_u(y)$, $u = 1, \ldots, U$. However, we stress that even for one target type $\tilde{u}$, $p_{\tilde{u}}(y)$ can depend on *all* types of events in the history through the CIF (Eq. 1 and 4).

In our model, for each target type $\tilde{u}$, the CIF $\lambda_{\tilde{u}}(t)$ is a transformation of a function $f$ drawn from a GP with mean $\mu$ and covariance function $K$, $f \sim \mathcal{GP}(\mu, K)$. We note that $f$, $\mu$ and $K$ can be different for $\lambda_u(t)$ of a different type $u$. The input of $f$ consists of the elapsed times since the last $Q$ events of all the types. That is, $f : \mathcal{X} \to \mathbb{R}$, where $\mathcal{X} \subseteq \mathbb{R}_{\geq 0}^D$, $D = U \times Q$, and $U$ is the size of the label set. To convert it to a valid CIF, we use the square transformation

$$\lambda_{\tilde{u}}(t) = f(x(t))^2 = f(t - s_1^1(t), \ldots, t - s_U^Q(t))^2 \tag{4}$$

where $s_u^q(t)$ is the time of the $q$-th (from last) event of type $u$ before time $t$, which could be undefined when no such event exists. The input of the GP is $x(t) = (t - s_u^q(t))_{u=1,q=1}^{U,Q} \in \mathcal{X}$. That is, $x$ depends on the current time $t$ and the last $Q$ events of *all* types. The $d$-th dimension of $x(t)$ is $x_d(t) = t - s_u^q(t)$, corresponding to the time elapsed since the $q$-th (from last) event of type $u$. In fact, $Q$ *does not have to be the same for each type* $u$, i.e., we can have a different $Q_u$ for each type $u$, but for notational simplicity, we use $Q$ as if it were the same for all types. We note that the CIF of the model *directly* depends on the past events and call the model a *GP regressive point process* (GPRPP).

The square transformation ensures nonnegativity of $\lambda$ and enables closed-form evaluation of the integrals in the likelihood as shown in the next section. It was originally proposed in [Lloyd et al., 2015] for GP-modulated point processes for event sequences without types and then exploited in later works (e.g, [Lloyd et al., 2016, Ding et al., 2018]). Compared with these works, where the GP is a function of the *single* "absolute" time, in our model, the GP is a function of *multiple* "relative" times since past events, which keeps changing as new events happen. This makes the inference much harder, and new efficient algorithms to evaluate the integrals are developed in this work.

A key challenge in the model definition is to deal with undefined inputs. That is, inevitably at some time $t$ (e.g., at the very beginning of a sequence), the $q$-th (from last) event of type $u$ may not exist. Inspired by Hawkes processes, we come up with a novel kernel for the GP. We start by augmenting each input $x_d(t)$ with an additional indicator $\mathbb{I}[x_d(t)]$ to indicate whether the $q$-th (from last) event of type $u$ exists, i.e., whether $s_u^q(t)$ (correspondingly $x_d(t)$) is defined. When $s_u^q(t)$ is undefined (there are less than $q$ events of type $u$ in the past), we can set a dummy value for $x_d(t)$ (e.g., $x_d(t) = \infty$), and $\mathbb{I}[x_d(t)] = 0$. Otherwise, $x_d(t) = t - s_u^q(t)$ as before, and $\mathbb{I}[x_d(t)] = 1$. The dimensionality of

the input essentially becomes $2D$, but for notational simplicity, the indicators are implicit in $x(t)$. We define the kernel as

$$K(x(t), x'(t')) = \sum_{d=1}^{D} \underbrace{\mathbb{I}\left[x_d(t)\right] \mathbb{I}\left[x'_d(t')\right]}_{K_1} \underbrace{\gamma_d \exp\left(-\frac{(x_d(t) - x'_d(t'))^2}{2\alpha_d}\right)}_{K_2} \tag{5}$$

where $\gamma_d, \alpha_d > 0$. This is essentially a sum of $D$ kernels, each of which is a product of two kernels $K_1$ and $K_2$. $K_2$ is the squared-exponential kernel on the value of $x_d(t)$. $K_1$ is the inner product on the indicator $\mathbb{I}\left[x_d(t)\right]$. We use the squared-exponential kernel, because it is widely used and has closed-form evaluations of $\psi$ and $\Psi$ as shown in the next section, but it can be replaced by any kernel with the latter property.

**Remark 1.** The two inputs of the kernel have different notations for $x$ and $t$, indicating they can come from different sequences at different *absolute* times. The kernel is actually isolated from the absolute time $t$ in the individual sequences, since it only depends on the value of $x(\cdot)$ at $t$. This is very different from previous GP-based models (e.g., [Lloyd et al., 2015, 2016, Ding et al., 2018]), where the inputs of the kernel always come from the same sequence, and the kernel depends on the absolute time $t$ in the sequence.

We make the following two assumptions to justify the definition of our model. A proof of Theorem 1 is in the supplementary material.

**Assumption 1.** *For each type $u$ of events, they have time-limited influences on the target events. That is, there is a time limit, $\Delta T_u < \infty$, for each type $u$ of event, such that for any event of type $u$ occurring at $s_u$, $\lambda_{\tilde{u}}(t)$ may depend on $s_u$ only if $0 < t - s_u \leq \Delta T_u$.*

**Assumption 2.** *For each type $u$ of events, there exists $M_u : \mathbb{R} \to \mathbb{Z}$ such that for any bounded time interval $\mathcal{I} = [t_{beg}, t_{end})$, $|\mathcal{I}| = t_{end} - t_{beg} < \infty$, the number of events $N_u(\mathcal{I}) \leq M_u(|\mathcal{I}|) < \infty$.*

**Theorem 1.** *Given that assumption 1 and 2 hold, there exists $Q < \infty$ such that $\lambda_{\tilde{u}}(t)$ depends on at most the last $Q$ events of any type at any time $t$.*

## 4 Conditional GPRPP

In this section, we propose a conditional GP model for GPRPP and call it conditional GP regressive point process (CGPRPP). The input of the GP is defined in the previous section and denoted as $x = x(t)$. When $t$ is not important or clear from the context, we just denote the input as $x$.

Previously, different forms of sparse GPs based on inducing variables have been proposed to improve the efficiency of GPs (e.g., [Quiñonero-Candela and Rasmussen, 2005, Titsias, 2009]). Typically, a set of inducing points are introduced in the input space, and the inducing variables corresponding to the points are *marginalized* out for learning and inference. Our idea is similar, but the difference is that we *condition* on the inducing variables, which correspond to the values of the CIF given different situations of the history. Therefore, we call these points *conditional points*.

Let $Z \in \mathcal{X}^M$ be a sequence of $M$ conditional points in the input space. We will explain how to pick these points later. Given any input $x \in \mathcal{X}$, the output of the GP is $f_x = f(x) \in \mathbb{R}$. Let $f_Z = f(Z) \in \mathbb{R}^M$. We define $\mu_x$ and $\mu_Z$ as the prior mean $\mu$ of the appropriate dimensions for $x$ and $Z$ respectively. Let $K_{xx'} = K(x, x')$ as defined in Eq. 5 for any inputs $x$ and $x'$. If $x$ and $x'$ are vectors, $K_{xx'}$ is the Gram matrix of the corresponding size. Then $p(f_Z) = \mathcal{N}(\mu_Z, K_{ZZ})$, and $p(f_x|f_Z) = \mathcal{N}(\mu_{x|Z}, \sigma^2_{x|Z})$, where

$$\mu_{x|Z} = \mu_x + K_{xZ}K_{ZZ}^{-1}(f_Z - \mu_Z), \qquad \sigma^2_{x|Z} = K_{xx} - K_{xZ}K_{ZZ}^{-1}K_{Zx} \tag{6}$$

From Eq. 3 and 4, the conditional density of the sequence $y$ given $f_x$ is

$$\ln p_{\tilde{u}}(y|f_x) = \sum_{n=1}^{N} \delta(u_n, \tilde{u}) \ln f(x(t_n))^2 - \int_0^T f(x(t))^2 dt \tag{7}$$

where $N = |y|$ is the total number of *all* types of events in $y$.

**Remark 2.** It is worth noting that the conditional points in $Z$ are independent of any specific sequence $y$, since they are points in $\mathcal{X}$.

Assuming we observe $f_Z = m_Z$, we can maximize the conditional density $p_{\tilde{u}}(y|m_Z)$ to learn the hyper-parameters of the model:

$$\ln p_{\tilde{u}}(y|m_Z) = \ln \int p_{\tilde{u}}(y|f_x)p(f_x|m_Z)df_x$$

where $p(f_x|m_Z)$ is defined by Eq. 6 with $f_Z = m_Z$. Because there is a correspondence between $f(x(t))$ and the CIF $\lambda(t)$, $m_Z$ essentially corresponds to different values of the CIF given different situations of the history, determined by different $z \in Z$. Even given the exact same history, the CIF may still be stochastic. Therefore, we allow noise in $f_Z$, which is a generalization of the noiseless case, so $f_Z = m_Z + \epsilon_Z$, where $p(\epsilon_Z) = \mathcal{N}(0, S_\epsilon)$. Then we can marginalize out $\epsilon_Z$ by integrating w.r.t. $p(\epsilon_z)$. In the end, we maximize

$$\ln p_{\tilde{u}}(y|m_Z) = \ln \iint p_{\tilde{u}}(y|f_x)p(f_x|m_Z,\epsilon_Z)p(\epsilon_Z)df_x d\epsilon_Z = \ln \int p_{\tilde{u}}(y|f_x)p(f_x|m_Z)df_x \quad (8)$$

where $p(f_x|m_Z,\epsilon_Z)$ is defined by Eq. 6, and

$$p(f_x|m_Z) = \int p(f_x|m_Z,\epsilon_Z)p(\epsilon_Z)d\epsilon_Z = \mathcal{N}(\tilde{\mu}_x, \tilde{\sigma}_x^2)$$

has a closed-form solution

$$\tilde{\mu}_x = \mu_x + K_{xZ}K_{ZZ}^{-1}(m_Z - \mu_Z), \quad \tilde{\sigma}_x^2 = K_{xx} - K_{xZ}K_{ZZ}^{-1}K_{Zx} + K_{xZ}K_{ZZ}^{-1}S_\epsilon K_{ZZ}^{-1}K_{Zx}. \quad (9)$$

**Remark 3.** Because we *condition* on the pseudo-observations $m_Z$, they can move freely when we optimize Eq. 8, and their values are determined by fitting to the training data. Intuitively, they act as key points of the CIF, which are supposed to capture the key information regarding the entire CIF.

Eq. 8 is hard to maximize directly. Instead, we derive a lower bound using Jensen's inequality and maximize the lower bound

$$\ln \int p_{\tilde{u}}(y|f_x)p(f_x|m_Z)df_x = \ln \mathbb{E}\left[p_{\tilde{u}}(y|f_x)\right] \geq \mathbb{E}\left[\ln p_{\tilde{u}}(y|f_x)\right] \quad (10)$$

where the expectation is w.r.t. $p(f_x|m_Z)$, and from Eq. 7

$$\mathbb{E}\left[\ln p_{\tilde{u}}(y|f_x)\right] = \sum_{n=1}^{N} \delta(u_n,\tilde{u})\mathbb{E}\left[\ln f(x(t_n))^2\right] - \sum_{n=1}^{N+1} \int_{t_{n-1}}^{t_n} \left(\mathbb{E}\left[f(x(t))\right]^2 + \text{Var}\left[f(x(t))\right]\right) dt$$

where we define $t_0 = 0$ and $t_{N+1} = T$ to be the start and end time of the sequence $y$.

From [Lloyd et al., 2015], we have

$$\mathbb{E}\left[\ln f(x(t_n))^2\right] = -\tilde{G}\left(-\frac{\tilde{\mu}_x^2}{2\tilde{\sigma}_x^2}\right) + \ln\left(\frac{\tilde{\sigma}_x^2}{2}\right) - C \quad (11)$$

where $C \approx 0.57721566$ is the Euler-Mascheroni constant, $\tilde{G}$ is defined via the confluent hypergeometric function, and $\tilde{\mu}_x^2 = \mathbb{E}\left[f_x\right]^2, \tilde{\sigma}_x^2 = \text{Var}\left[f_x\right]$ can be computed as in Eq. 9.

Meanwhile,

$$\int_{t_{n-1}}^{t_n} \mathbb{E}\left[f_x\right]^2 dt = (t_n - t_{n-1})\mu_x^2 + 2\mu_x \psi_n^T K_{ZZ}^{-1}(m_Z - \mu_Z)$$
$$+ (m_Z - \mu_Z)^T K_{ZZ}^{-1}\Psi_n K_{ZZ}^{-1}(m_Z - \mu_Z), \quad (12)$$

$$\int_{t_{n-1}}^{t_n} \text{Var}\left[f_x\right] dt = \sum_{d=1}^{D} \int_{t_{n-1}}^{t_n} \gamma_d \mathbb{I}\left[x_d(t)\right] dt - \text{Tr}\left(K_{ZZ}^{-1}\Psi_n\right) + \text{Tr}\left(K_{ZZ}^{-1}S_\epsilon K_{ZZ}^{-1}\Psi_n\right). \quad (13)$$

The definitions of $\psi$ and $\Psi$ are complex and included in the supplementary material. We note that the computation of them is the bottleneck of the learning algorithm. A straightforward algorithm would cost $O(M^2ND^2)$. However, by merging computation related to the same event type, we can reduce the complexity to $O(M^2NDQ)$. By setting each conditional point active on only one dimension, we can reduce the complexity to $O(M^2NQ)$. If we further set $Q_{\tilde{u}} > 1$ only for the target type and $Q_u = 1$ for $u \neq \tilde{u}$, and assume $N_{\tilde{u}}Q_{\tilde{u}} = O(N)$, where $N_u$ is the number of points of type $u$, the complexity can be reduced to $O(M^2N)$, which is what we adopt in the experiments. The details and a proof are in the supplementary material.

**Learning**   We also add an independent noise kernel $\sigma^2 I$ to the existing kernel (Eq. 5), which results in a new term in the integral of the variance (Eq. 13). For learning the model, we maximize the lower bound (Eq. 10) w.r.t. the set of hyper-parameters $\Theta = \{\mu, \alpha, \gamma, \sigma, m_Z, S_\epsilon\}$.

We assume that $m_Z$ provides sufficient information for the inference on the test data $\mathcal{D}_*$.

**Assumption 3.** *Conditioned on $m_Z$, the test data $\mathcal{D}_*$ is independent from the training data $\mathcal{D}$,* $p(\mathcal{D}_*|\mathcal{D}, m_Z) = p(\mathcal{D}_*|m_Z)$.

**Inference**   For inference of the test likelihood, to compare with non-Bayesian models, we use a point estimate of the CIF, instead of model averaging. We use the optimal hyper-parameters $\Theta^*$ learned from the training data $\mathcal{D}$ to estimate the mean CIF

$$\lambda_{\tilde{u}}^*(t) = \mathbb{E}\left[\lambda_{\tilde{u}}(t)|\Theta^*\right] = \mathbb{E}\left[f(x(t))|\Theta^*\right]^2 + \text{Var}\left[f(x(t))|\Theta^*\right] \tag{14}$$

on the test data $\mathcal{D}_*$, where the conditional mean and variance are defined in Eq. 9. Then we use the mean CIF as our prediction to compute the likelihood $p(\mathcal{D}_*|\lambda^*)$ on the test data. For prediction of event times, a sampling-based algorithm is derived in the supplementary material.

**Conditional point placement**   Due to the high dimensionality of the input space of $f$, it is preferable that the conditional points are placed beforehand and fixed in the learning procedure. Based on the additive form of our kernel, we place the conditional points independently on each dimension. Each conditional point will be active on only one dimension. In our experiments, for simplicity, we put the conditional points regularly on each dimension within a region. If prior knowledge is available, it can be used to determine the region; otherwise, we can use the following heuristics. The left bound of the region is usually 0. The right bound can be set to the maximum (or some quantile) of the time span between two ($Q = 1$) or more ($Q > 1$) consecutive points of the same type, since beyond that, the conditional points will have limited effects.

## 5   Experiments

We compare our method with two state-of-the-art nonparametric Hawkes process variants. **HP-GS** [Xu et al., 2016] is a nonparametric Hawkes process using a set of (Gaussian) basis functions to approximate the triggering kernels, with sparse and group lasso regularization. For each experiment, we tune its tuning parameters $\alpha_S$ and $\alpha_G$ in a wide range $\{10^{-2}, 10^{-1}, \dots, 10^4\}$ as in the original work using cross-validation based on the likelihood. In all the experiments, the bandwidth of the Gaussian kernels is set to be optimal, that is the inverse of the cut-off frequency, based on the positions of the kernels. The cut-off frequency $\omega_0 = \pi M/T$, where $M$ is the number of kernels and $T$ is the right bound on the kernels. **HP-LS** [Eichler et al., 2017] is another nonparametric Hawkes process. This method allows very flexible triggering kernels to be estimated by discretizing the kernels and solving a least-square problem. Its parameters are set in accordance with the other methods for each experiment. For our method, to improve efficiency, when we set $Q > 1$, we only set it for the target type and keep $Q = 1$ for the others, as discussed in the supplementary material. We tie the parameters for different dimensions $q = 1, \dots, Q$ of the same type $u$.

### 5.1   Synthetic datasets

First we generate two synthetic datasets representing two distinctive types of event sequences using the thinning algorithm [Ogata, 1981]. The first dataset is generated through a renewal process. The baseline intensity is $\mu$. When there is a new event, the intensity temporarily becomes $A(1 - \sin(2\pi t/\tau))$, for a limited time $t \in (0, \tau)$ after the event. Each new event will reset the intensity. We set $\mu = 0.1, A = 0.1, \tau = 20$.

The second dataset is generated through a Hawkes process. The baseline intensity is $\mu$. The triggering kernel is $A \exp(-(t - b)^2/\sigma^2)$, i.e., a Gaussian kernel. Different from the renewal process, each new event will add a new Gaussian kernel on top of the existing intensity. We set $\mu = 0.1, A = 0.1, b = 10, \sigma^2 = 4$.

For each dataset we generate 200 sequences of length of 100 time units each. Each dataset is split into 100 training sequences and 100 testing sequences. For the first dataset, we set $Q = 1$ and conditional points at $0, 5, \dots, 15$ for CGPRPP. For HP-GS, the kernels are also placed at $0, 5, \dots, 15$. For HP-LS,

we set $h = 1, k = 20$. For the second dataset, we use the same settings for all the methods, except that we vary $Q = 1, 5, 10, 20, 40$ for CGPRPP to see the effect of adding more regression terms.

We visualize the influence from a past event of a specific type to the target events as the changes in the intensity of the target type over time since that event. For Hawkes processes, it is similar to plotting the triggering kernels, except that the triggering kernels are added on top of the baseline intensity, so we can compare the intensity after an event with the baseline intensity. For CGPRPP, it is equivalent to simulating an event at time 0 and plotting the changes in the intensity as time elapses.

The true influence functions are in the first column in Figure 2, followed by the inferred influence functions for each method. For the first dataset, HP-GS cannot learn the influence function, because its limitation in the dependencies of the CIF on the past events. Although the triggering kernels are nonparametric, the baseline intensity and the triggering kernels are additive in the CIF. This limitation is quite common in nonparametric Hawkes processes (e.g., [Zhang et al., 2018, Donnet et al., 2018, Zhou et al., 2013b]). HP-LS is more flexible and learns a better influence function, but the discretization tends to make the function noisy. CGPRPP almost completely recovers the true influence function. We note that this influence represents an inhibition followed by an excitation, which is common in practice such as neural spike trains [Eichler et al., 2017]. However, most Hawkes process variants can only model either excitations or inhibitions, but not a mix of both at the same time. In contrast, CGPRPP models the whole CIF as a nonparametric function of the past events and therefore can model these more complex dependencies.

For the second dataset, HP-GS is a perfect match for the data, so unsurprisingly it recovers the influence function very well. HP-LS also learns the influence reasonably well, although still suffering from discretization. Interestingly, CGPRPP with $Q = 40$ (similar for $Q = 10, 20$) learns an influence function very close to HP-GS.

The test log-likelihood on the synthetic datasets are shown in Table 1. CGPRPP performs the best on the first dataset, while HP-GS and CGPRPP perform similarly on the second dataset, with CGPRPP being marginally better. The likelihood results are concordant with how well the models recovered the influence function. For the second dataset, we show the performance of CGPRPP selected with the *training* likelihood ($Q = 40$). A comparison of GPRPP based on variational sparse GP [Lloyd et al., 2015] and conditional GP, and the effect of varying $Q$ on the performance of CGPRPP are in the supplementary material.

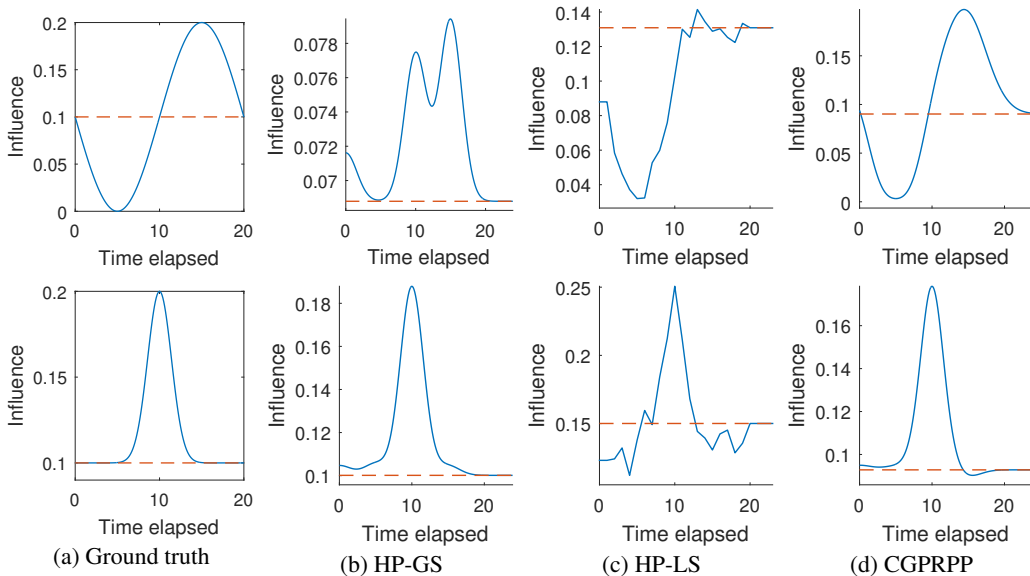

(a) Ground truth     (b) HP-GS     (c) HP-LS     (d) CGPRPP

Figure 2: Influences from past events on the first (top) and second (bottom) synthetic datasets. Solid lines are the CIFs after an event. Dashed lines are the baseline intensities. The ground truth is in the first column, followed by the result of each method.

Table 1: Test log-likelihood on synthetic datasets.

| Data | HP-GS | HP-LS | CGPRPP |
|---|---|---|---|
| 1 | -2671 | -2770 | **-2455** |
| 2 | -4074 | -4161 | **-4071** |

Table 2: Test log-likelihood on IPTV dataset.

| Month | HP-GS | HP-LS | CGPRPP |
|---|---|---|---|
| 1 | **-1.477e+05** | -1.779e+05 | -1.479e+05 |
| 2 | -1.509e+05 | -1.825e+05 | **-1.502e+05** |
| 3 | **-1.608e+05** | -1.928e+05 | -1.608e+05 |

## 5.2 IPTV dataset

The IPTV dataset consists of TV viewing records of users over 11 months [Luo et al., 2014, Xu et al., 2016]. Each sequence consists of times and types of the TV programs viewed by a user. Events in this dataset are generally very bursty, i.e., one event tends to trigger a group of events of the same type happening in a relatively short amount of time, while the distance between these burst groups are relatively large. This is a distinctive characteristic of data generated by Hawkes processes, so we expect the Hawkes process baselines to perform well. To the best of our knowledge, HP-GS has the best performance on this dataset, but our goal is to confirm whether CGPRPP can also fit the data well and achieve similar or better performance.

The data are extracted from THAP[1] [Xu and Zha, 2017], which contain 302 users in total. For efficiency, we randomly sample 200 users and use 100 for training and the others for testing. All the models are trained on 1 month and tested on the following 3 months. More details are in the supplementary material. For HP-GS, we put the kernels at every 20 minutes from 0 up to 24 hours, since the length of most TV programs is about 20 to 40 minutes [Xu et al., 2016]. For HP-LS, we train multiple models with $h = 1.25, 5, 20$ minutes and $k = (24 * 60 + 20)/h$ correspondingly. For CGPRPP, the conditional points are also placed at every 20 minutes up to 24 hours. We set $Q = 5, 10, 20$ and select $Q$ based on the *training* likelihood.

Table 2 shows the test log-likelihood of the models on different months. This is the total log-likelihood of all types of events. For HP-LS, we show the best *test* log-likelihood across different $h$ and $k$. As expected, HP-GS performs the best, confirming the bursty characteristic of the data. However, HP-LS does not perform well. A problem of HP-LS is that the discretization tends to make the influence function noisy and fail to generalize well. CGPRPP has a competitive performance close to HP-GS, showing its capability to model bursty events.

## 5.3 MIMIC datasets

To show the flexibility of CGPRPP in modeling other complex event patterns than the bursty patterns as in many previously used datasets similar to the IPTV dataset, we derive multiple new event sequence datasets from MIMIC III [Johnson et al., 2016] consisting of lab tests ordered to patients in a hospital. Lab orders tend to have more complex dependencies such as a complex mix of multiple inhibitions and excitations over time (e.g., see Figure 3).

Since there are labs that tend to occur together, we group them into several lab classes. We extract 20 different datasets targeting the most frequent 20 classes. Each dataset consists of 10 different lab classes, one of which is the target we try to predict, while the others are the predictors. We sample 200 admissions (sequences) randomly from each dataset, where 100 admissions are used for training and the others for testing. More details are in the supplementary material.

For HP-GS, we put the kernels at $0, 8, \ldots, 48$ hours. We also test a different version of the method, HP-GS-A, using the adaptive basis-function-selection algorithm in [Xu et al., 2016] to place the kernels. For HP-LS, we train multiple models with $h = 0.5, 2, 8$ hours and $k = (48 + 8)/h$ correspondingly. For CGPRPP, the conditional points are also placed at $0, 8, \ldots, 48$ hours. We set $Q = 1, 10$ and select $Q$ based on the *training* likelihood. As a reference, we also test against a model based on deep neural networks, the neural self-modulating multivariate point process (NSMMPP) [Mei and Eisner, 2017]. The number of hidden units is selected from $64, 128, \ldots, 1024$ as in the original work through a validation set (80/20 split from the full training set).

The test log-likelihood of the models is shown in Table 3. Each column is a different dataset with a different target lab class. They are ordered from the most frequent (355) to the least frequent (18) based on their occurrences. For HP-LS, we show the best *test* log-likelihood across different $h$ and $k$ on each dataset. CGPRPP achieves the best or close to the best performance on all datasets except class 550 and 18. On class 355, 60, 151, 113, and 140, CGPRPP outperforms the second best by a large margin. In some cases (e.g., class 550) CGPRPP with a different Q actually has a much better result, although not being selected. We also conducted time prediction experiments to predict the time for each target event. The results also show the advantage of CGPRPP. The full likelihood and time prediction results are in the supplementary material.

As an example, we plot the influence functions for class 355 from past events of the same type in Figure 3. HP-GS learns a smooth influence function with excitations around 24 and 48 hours. This corresponds to the fact that right after a lab being ordered, it might need to be repeated after one or two days. In contrast, HP-LS ($h = 0.5$ with the best test likelihood) learns a much noisier pattern due to discretization, which is harder to interpret. Compared with HP-GS, CGPRPP learns not only similar excitations around 24 and 48 hours, but also a strong inhibition after each excitation, showing a more flexible fit to the data.

Table 3: Test log-likelihood on MIMIC lab order datasets.

| Dataset | 355 | 60 | 3 | 95 | 368 | 354 | 151 | 550 | 113 | 140 |
|---------|-----|-----|-----|-----|-----|-----|-----|-----|-----|-----|
| HP-GS | -3668 | -4673 | **-3721** | -4064 | -3366 | -4344 | -3338 | -1053 | -4656 | -3206 |
| HP-GS-A | -3947 | -5051 | -3733 | -4390 | -3711 | -4792 | -3574 | -1064 | -5049 | -3475 |
| HP-LS | -6510 | -7299 | -5722 | -5712 | -5625 | -7185 | -5323 | -1744 | -7143 | -4625 |
| NSMMPP | -3664 | -4660 | -3737 | -3982 | **-3309** | -4409 | -3763 | **-1039** | -4539 | -3244 |
| CGPRPP | **-3249** | **-4246** | -3759 | **-3933** | -3378 | **-4225** | **-3093** | -1175 | **-4276** | **-2942** |

| Dataset | 294 | 17 | 150 | 80 | 394 | 1 | 53 | 7 | 8 | 18 |
|---------|-----|-----|-----|-----|-----|-----|-----|-----|-----|-----|
| HP-GS | -1011 | -3783 | -3238 | **-3388** | -3098 | **-3220** | -1913 | **-2502** | **-1633** | -1596 |
| HP-GS-A | -1054 | -3807 | -3537 | -3772 | -3251 | -3291 | -2138 | -2533 | -1667 | -1678 |
| HP-LS | -1308 | -5339 | -4894 | -5365 | -4945 | -3772 | -2963 | -3514 | -3142 | -3085 |
| NSMMPP | **-941.2** | **-3758** | -3377 | -3903 | -3268 | -3228 | -1916 | -2626 | -1786 | **-1532** |
| CGPRPP | -993.6 | -3808 | **-3100** | -3402 | **-3010** | -3234 | **-1900** | -2512 | -1694 | -1648 |

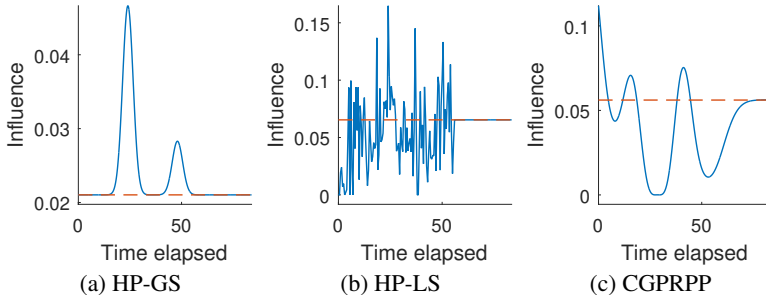

(a) HP-GS          (b) HP-LS          (c) CGPRPP

Figure 3: Influences from past events of the same type as the target class 355 on the MIMIC dataset.

## 6   Conclusion

In this work, we proposed a new nonparametric method for modeling dependencies between events in event sequences using Gaussian processes. Similar to Hawkes processes and different from previous GP-modulated point processes, the proposed model can be learned on a sample of sequences and then applied to other unseen sequences. However, we showed that the proposed model is more flexible than state-of-the-art nonparametric Hawkes process variants. It can learn the dependencies between events that are common in practice but difficult for the Hawkes process variants to represent, e.g., a mix of inhibitions and excitations after an event. Our method showed competitive or better performance on different datasets compared with the Hawkes process variants.

## Acknowledgement

This work was supported by NIH grant R01-GM088224. Siqi Liu was also supported by CS50 Merit Pre-doctoral Fellowship from the Department of Computer Science, University of Pittsburgh. The content of this paper is solely the responsibility of the authors and does not necessarily represent the official views of the NIH.

## Footnotes

[1]`https://github.com/HongtengXu/Hawkes-Process-Toolkit`

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
