[Supplementary Material · supplementary.pdf]

# (Supplementary Material) Nonparametric Regressive Point Processes Based on Conditional Gaussian Processes

**Siqi Liu**
Department of Computer Science
University of Pittsburgh
Pittsburgh, PA 15213
siqiliu@cs.pitt.edu

**Milos Hauskrecht**
Department of Computer Science
University of Pittsburgh
Pittsburgh, PA 15213
milos@pitt.edu

## A  Proof of Theorem 1

*Proof.* Given any event sequence $y = \{(t_i, u_i)\}_i^{|y|}$, the CIF $\lambda_{\tilde{u}}(t)$ at any time $t$ may only depend on the subset of events $\{(t_i, u_i) \in y : 0 < t - t_i \leq \Delta T_{u_i}\}$, according to Assumption 1. Focusing on a specific type $u$, the subset of events that $\lambda_{\tilde{u}}(t)$ may depend on is $\{(t_i, u_i) \in y : 0 < t - t_i \leq \Delta T_u, u_i = u\}$. Notice that all of these events occur within the time interval $[t - \Delta T_u, t)$, which is a bounded interval, since $\Delta T_u < \infty$. Therefore, from Assumption 2, we have $N_u([t - \Delta T_u, t)) \leq M_u(\Delta T_u) < \infty$ for some $M_u$, which holds for any time $t$. That is, $\lambda_{\tilde{u}}(t)$ depends on at most the last $M_u(\Delta T_u)$ events of type $u$ at any time $t$. To complete the proof, let $Q = \max_{u=1,\ldots,U} M_u(\Delta T_u)$. $\qquad\square$

## B  Definitions of $\psi$ and $\Psi$

Define $v_{d,n,m} = t_n - s_d(t_m)$, where $s_d(t)$ is the $q_d$-th (from last) point of type $u_d$ before $t$ ($q_d$ and $u_d$ are determined by the dimension $d$). Then for any $z, z' \in Z$

$$\psi_n(z) = \sum_{d=1}^{D} \mathbb{I}[z_d] \mathbb{I}[s_d(t_n)] \gamma_d \frac{\sqrt{\pi \alpha_d}}{\sqrt{2}} \left[ \mathrm{erf}\left( \frac{v_{d,n,n} - z_d}{\sqrt{2\alpha_d}} \right) - \mathrm{erf}\left( \frac{v_{d,n-1,n} - z_d}{\sqrt{2\alpha_d}} \right) \right] \quad (15)$$

$$\Psi_n(z, z') = \sum_{i,j}^{D} \mathbb{I}[z_i] \mathbb{I}[z_j'] \mathbb{I}[s_i(t_n)] \mathbb{I}[s_j(t_n)] \gamma_i \gamma_j \frac{\sqrt{\pi \alpha_i \alpha_j}}{\sqrt{2(\alpha_i + \alpha_j)}}$$

$$\exp\left( -\frac{(z_i + s_i(t_n) - z_j' - s_j(t_n))^2}{2(\alpha_i + \alpha_j)} \right)$$

$$\left[ \mathrm{erf}\left( \frac{\alpha_i(v_{j,n,n} - z_j') + \alpha_j(v_{i,n,n} - z_i)}{\sqrt{2\alpha_i \alpha_j(\alpha_i + \alpha_j)}} \right) \right.$$

$$\left. -\mathrm{erf}\left( \frac{\alpha_i(v_{j,n-1,n} - z_j') + \alpha_j(v_{i,n-1,n} - z_i)}{\sqrt{2\alpha_i \alpha_j(\alpha_i + \alpha_j)}} \right) \right] \quad (16)$$

## C  Efficient computation of $\psi$ and $\Psi$

We note that both $\sum_n \psi_n$ and $\sum_n \Psi_n$ can be combined to improve efficiency. A straightforward implementation would cost $O(ND^2)$, where $N$ is the total number of points. The key thing to notice

is that for fixed dimensions $d, i, j$, the types of points that matter are only the ones related to the dimensions, while we can integrate over the other types of points in closed form. Specifically,

$$\sum_n \psi_n(z) = \sum_{d=1}^{D} \mathbb{I}\left[z_d\right] \gamma_d \frac{\sqrt{\pi \alpha_d}}{\sqrt{2}} g_d \tag{17}$$

$$\sum_n \Psi_n(z, z') = \sum_{i,j}^{D} \mathbb{I}\left[z_i\right] \mathbb{I}\left[z'_j\right] \gamma_i \gamma_j \frac{\sqrt{\pi \alpha_i \alpha_j}}{\sqrt{2(\alpha_i + \alpha_j)}} G_{ij} \tag{18}$$

where

$$g_d = \sum_{k=b_d}^{N_{u_d}+1} \left[ \mathrm{erf}\left( \frac{v_{d,(k),(k)} - z_d}{\sqrt{2\alpha_d}} \right) - \mathrm{erf}\left( \frac{v_{d,(k-1),(k)} - z_d}{\sqrt{2\alpha_d}} \right) \right]$$

$$G_{ij} = \sum_{k=b_{ij}}^{N_{u_i,u_j}+1} \exp\left( -\frac{(z_i + s_i(t_{(k)}) - z'_j - s_j(t_{(k)}))^2}{2(\alpha_i + \alpha_j)} \right)$$

$$\left[ \mathrm{erf}\left( \frac{\alpha_i(v_{j,(k),(k)} - z'_j) + \alpha_j(v_{i,(k),(k)} - z_i)}{\sqrt{2\alpha_i \alpha_j(\alpha_i + \alpha_j)}} \right) \right.$$

$$\left. -\mathrm{erf}\left( \frac{\alpha_i(v_{j,(k-1),(k)} - z'_j) + \alpha_j(v_{i,(k-1),(k)} - z_i)}{\sqrt{2\alpha_i \alpha_j(\alpha_i + \alpha_j)}} \right) \right]$$

For $g_d$, $N_{u_d}$ is the number of points of type $u_d$, $t_{(k)}$ is the time of the $k$-th such point (i.e., $(k)$ maps the index $k$ in the sub-sequence of type $u_d$ to the index of the same point in the full sequence), $b_d$ is the index of the first such point with at least $q_d$ points of type $u_d$ before it, and $t_{(N_{u_d}+1)} = T$. For $G_{ij}$, $N_{u_i,u_j} = N_{u_i} + N_{u_j}$ is the number of points in the combined sequence of points of both type $u_i$ and type $u_j$, $t_{(k)}$ is the time of the $k$-th such point, $b_{ij}$ is the index of the first such point with at least $q_i$ points of type $u_i$ and $q_j$ points of type $u_j$ before it, and $t_{(N_{u_i,u_j}+1)} = T$.

In this way, the calculation of $\sum_n \psi_n(z)$ and $\sum_n \Psi_n(z, z')$ can be done in $O(NDQ)$ if we share the same $Q$ across all types $u$. If we only set $Q > 1$ for one type, e.g., for $u = 1$, and set $Q = 1$ for the other types, and if $N_1 Q_1 = O(N)$, then the bound becomes $O(ND)$. Either way, it is an improvement compared with $O(ND^2)$ for a straightforward implementation. A proof is in the next section. Empirically, we can improve the performance even further by pre-calculating once and storing the values of $v$ and $s_i - s_j$ at the beginning.

## D   Proof of $\Psi$ computational complexity

*Proof.* We only prove the bound for $\sum_n \Psi_n(z, z')$, since $\sum_n \psi_n(z)$ is more efficient to compute. To compute $\sum_n \Psi_n(z, z')$, we need to sum over all pairs of dimensions $i, j = 1, \ldots, D$. However, for each pair of $(i, j)$, we only need to sum over at most $N_{u_i} + N_{u_j}$ items, where $N_u$ is the number of points of type $u$, and $u_i, u_j$ are the types for dimension $i, j$. The reason is that the points of the other types in the middle can be integrated over in closed-form. Therefore, the total number of items to sum over is at most

$$\sum_{i=1}^{D} \sum_{j=1}^{D} (N_{u_i} + N_{u_j}) = 2D \sum_{u=1}^{U} N_u Q_u$$

where $Q_u$ is the regression hyper-parameter $Q$ for type $u$.

In summary, in the most general case, the complexity is $O(D \sum_u N_u Q_u)$. If we use the same $Q$ for all types, then it becomes $O(NDQ)$. If we use $Q > 1$ for only one type, say $u = 1$, and $N_1 Q_1 = O(N)$, then it becomes $O(ND)$. In practice, we can use the symmetric property of the sum and almost halve the amount of computation.

In the most general case when each conditional point can have $D$ active dimensions, we need to do the above computation for each pair of $(z, z')$. The total complexity is $O(M^2 NDQ)$, where $M$ is the total number of conditional points. However, if we set each conditional point active on only one dimension, then the complexity becomes $O(M^2 NQ)$. Additionally, if we set $Q > 1$ for only one type, say $u = 1$, and $N_1 Q_1 = O(N)$, then it becomes $O(M^2 N)$. $\qquad\square$

# E    Time prediction

For predicting the time of the next target event, given the history up to a time point $t$, we compute the *expected time* for the next target event given the CIF $\lambda_{\tilde{u}}$

$$\mathbb{E}\left[s_{\tilde{u}}|\lambda_{\tilde{u}}\right] = \int_t^\infty s_{\tilde{u}} \lambda_{\tilde{u}}(s_{\tilde{u}}) \exp\left(-\int_t^{s_{\tilde{u}}} \lambda_{\tilde{u}}(v)dv\right) ds_{\tilde{u}} \tag{19}$$

for $s_{\tilde{u}} \in (t, \infty)$, where $\lambda_{\tilde{u}}$ depends on the history $\mathcal{H}_t$ and $f_x$. That is

$$\mathbb{E}\left[s_{\tilde{u}}|\lambda_{\tilde{u}}\right] = \mathbb{E}\left[s_{\tilde{u}}|\mathcal{H}_t, f_x\right] = \int_t^\infty s_{\tilde{u}} f(x(s_{\tilde{u}}))^2 \exp\left(-\int_t^{s_{\tilde{u}}} f(x(v))^2 dv\right) ds_{\tilde{u}} \tag{20}$$

From here, we can take expectation w.r.t. $f_x$ using the conditional-point approximation. In the end, the prediction is

$$\begin{aligned}
\mathbb{E}\left[s_{\tilde{u}}|\mathcal{H}_t\right] &= \iint \mathbb{E}\left[s_{\tilde{u}}|\mathcal{H}_t, f_x\right] p(f_x|m_Z^*, \epsilon_Z) p(\epsilon_Z|S_\epsilon^*) df_x d\epsilon_Z \\
&= \int \mathbb{E}\left[s_{\tilde{u}}|\mathcal{H}_t, f_x\right] p(f_x|m_Z^*, S_\epsilon^*) df_x
\end{aligned} \tag{21}$$

where $m_Z^*$ and $S_\epsilon^*$ are part of the hyper-parameters $\Theta^*$ learned from the training data. The expectation w.r.t. $f_x$ is evaluated using Monte-Carlo sampling, and $\mathbb{E}\left[s|\mathcal{H}_t, f_x\right]$ is evaluated by sampling the point process through Ogata's modified thinning algorithm [Ogata, 1981].

An alternative approach, which is more efficient, is to use the mean CIF

$$\lambda_{\tilde{u}}^*(s_{\tilde{u}}) = \mathbb{E}\left[\lambda_{\tilde{u}}(s_{\tilde{u}})|\Theta^*\right] = \mathbb{E}\left[f(x(s_{\tilde{u}}))|\Theta^*\right]^2 + \text{Var}\left[f(x(s_{\tilde{u}}))|\Theta^*\right], \quad s_{\tilde{u}} \in (t, \infty) \tag{22}$$

to predict the events without sampling $f_x$. That is, we estimate $\lambda_{\tilde{u}}^*$ using the learned hyper-parameters $\Theta^*$ and the history $\mathcal{H}_t$, and plug $\lambda_{\tilde{u}}^*$ into Eq. 19 to estimate the time to the next event. We used this approach in the experiments and found it to be effective.

# F    Conditional GP vs. variational sparse GP

We compare the performance of GPRPP based on variational sparse GP with inducing points [Lloyd et al., 2015] and CGPRPP based on conditional GP with conditional points. Figure 4 shows the test log-likelihood of GPRPP and CGPRPP with $Q = 1, 5, 10, 20, 40$ on the second synthetic dataset. Conditional-GP-based model outperforms variational-sparse-GP-based model in all cases, showing that conditional GP can capture the dependencies between events better.

# G    Effect of varying $Q$

Figure 4 shows the test log-likelihood of CGPRPP with $Q = 1, 5, 10, 20, 40$ on the second synthetic dataset. We notice that $Q$ does affect the performance of CGPRPP, especially when it is small and the model is a mismatch for the data. However, as $Q$ increases, the performance tends to stabilize. For data generated through Hawkes processes, it is beneficial to have $Q$ large enough so the model is capable of approximating the compound influences from all the past events. However, in general, for data generated through processes other than Hawkes processes, an optimal $Q$ may need to be neither too small nor too large, and therefore selecting $Q$ may be necessary. In the experiments, we simply use the training likelihood to select $Q$, which turns out to be effective in most cases. A potential improvement is to use cross-validation, which we do not explore in this work.

# H    Details of the IPTV dataset

The IPTV dataset consists of TV viewing records of users over 11 months [Luo et al., 2014, Xu et al., 2016]. The dataset we use is extracted from THAP [Xu and Zha, 2017] that is publicly available[1].

Figure 4: The test log-likelihood of GPRPP based on variational sparse GP (GPRPP) and conditional GP (CGPRPP) with $Q = 1, 5, 10, 20, 40$.

The original dataset contains 302 users and 16 different types of events (genres of TV programs). Table 4 shows the counts of these different types of events. For efficiency, we randomly sampled 200 users and used 100 users for training and the others for testing. We removed the last two types of programs, "education" and "ads", due to extremely low counts.

We used data in March for training and the following months for testing (on separate users). We picked March a priori, because it has fewer irregularities such as holidays than the first two months.

Table 4: IPTV event types and counts.

| Type | Count |
|---|---|
| drama | 284092 |
| news | 190584 |
| entertainment | 122773 |
| others | 116449 |
| sports | 74502 |
| kids | 39712 |
| movie | 33437 |
| daily life | 33225 |
| economy | 23985 |
| law | 13636 |
| music | 12456 |
| documentary | 11162 |
| military | 10007 |
| science | 6790 |
| education | 798 |
| ads | 390 |

# I  Details of the MIMIC datasets

In the MIMIC dataset, there are labs that tend to occur together. We collect these labs into groups, which we call lab classes. These classes are built using the following procedure. First, we collect the occurrences of all the labs. Then, we calculate the Intersection over Union (IoU) for each pair of labs based on their occurrence timestamps. That is, if two labs always co-occur, then their IoU will be 1. In contrast, if they never co-occur, then it will be 0. Finally, we put two labs into the same class, if their IoU is above 0.95.

In the experiments, we focus on patients that have been admitted to the hospital. Within these patients, we have 710 types of labs. After grouping them, we get 598 classes. We pick the most frequent 20 classes as our targets, which are shown in Table 5. The labels of the labs in the same class are separated by semicolons. For each class, the labs all share the same property (without forcing it) in terms of "fluid" and "category", confirming that our grouping algorithm is reasonable.

To build the predictors for each target lab class, we find 10 different lab classes using heuristics. First, we find the admissions that have at least one occurrence of the target. Then, we calculate the

Table 5: Target lab classes used for experiments.

| Class ID | Lab labels | Fluid | Category | Count |
|---|---|---|---|---|
| 355 | Hemoglobin; MCH; MCHC; MCV; Platelet Count; RDW; Red Blood Cells; White Blood Cells | Blood | Hematology | 4619733 |
| 60 | Anion Gap; Bicarbonate; Chloride; Sodium | Blood | Chemistry | 2500535 |
| 3 | Base Excess; Calculated Total CO2; pCO2; pO2 | Blood | Blood Gas | 1942338 |
| 95 | Creatinine; Urea Nitrogen | Blood | Chemistry | 1236906 |
| 368 | INR(PT); PT | Blood | Hematology | 756797 |
| 354 | Hematocrit | Blood | Hematology | 693788 |
| 151 | Potassium | Blood | Chemistry | 669880 |
| 550 | Bilirubin; Blood; Glucose; Ketone; Leukocytes; Nitrite; Urine Appearance; Urine Color; Urobilinogen | Urine | Hematology | 598026 |
| 113 | Glucose | Blood | Chemistry | 595635 |
| 140 | Magnesium | Blood | Chemistry | 559517 |
| 294 | Basophils; Eosinophils; Lymphocytes; Monocytes; Neutrophils | Blood | Hematology | 547408 |
| 17 | pH | Blood | Blood Gas | 524600 |
| 150 | Phosphate | Blood | Chemistry | 489990 |
| 80 | Calcium, Total | Blood | Chemistry | 484701 |
| 394 | PTT | Blood | Hematology | 403567 |
| 1 | Specimen Type | Blood | Blood Gas | 398697 |
| 53 | Alanine Aminotransferase (ALT); Asparate Aminotransferase (AST) | Blood | Chemistry | 296876 |
| 7 | Free Calcium | Blood | Blood Gas | 246208 |
| 8 | Glucose | Blood | Blood Gas | 193253 |
| 18 | Potassium, Whole Blood | Blood | Blood Gas | 187020 |

event-wise probability of occurrence and admission-wise probability of occurrence for each class. The former is defined as the number of occurrences for the class divided by the total number of occurrences for all classes. The latter is defined as the number of admissions having at least one occurrence of the class divided by the total number of admissions. The difference between the two is that the former puts the frequency of the class over time into consideration, while the latter only considers the "popularity" of the class among the admissions.

After calculating the two probabilities, we keep only the classes that have an admission-wise probability greater than 0.5. Then we rank these classes by the ratio of the event-wise probabilities of occurrence between the subpopulation of admissions containing at least one target and the whole population, and pick the top 10. The intuition is that the latter probability can be seen as a prior probability of the event occurring, while the former as a posterior probability conditioned on that the target is present in the sequence (admission). Denote the event that, given a lab occurs, it is of the specific class $u$ as $E_u$, and the event that a lab of the target class $\tilde{u}$ also occurs in the same sequence as $O_{\tilde{u}}$. Then essentially, we iteratively find each predictor $u$ as

$$\arg \max_u \frac{p(E_u|O_{\tilde{u}})}{p(E_u)} = \arg \max_u \frac{p(E_u|O_{\tilde{u}})p(O_{\tilde{u}})}{p(E_u)} = \arg \max_u p(O_{\tilde{u}}|E_u).$$

Using the above heuristics, the target itself will always be selected as the top 1 predictor. Table 6 shows an example of the selected predictors for lab class 355. The first row is the target class itself, followed by the other predictors.

## J  Full likelihood results on MIMIC

Table 7 shows the full results of test log-likelihood on the MIMIC datasets. CGPRPP-1 and CGPRPP-10 are CGPRPP with $Q = 1$ and $Q = 10$. CGPRPP* is the model selected with the best training likelihood.

Table 6: Predictors selected for lab class 355.

| Class ID | Lab labels | Fluid | Category |
|---|---|---|---|
| 355 | Hemoglobin; MCH; MCHC; MCV; Platelet Count; RDW; Red Blood Cells; White Blood Cells | Blood | Hematology |
| 294 | Basophils; Eosinophils; Lymphocytes; Monocytes; Neutrophils | Blood | Hematology |
| 394 | PTT | Blood | Hematology |
| 368 | INR(PT); PT | Blood | Hematology |
| 140 | Magnesium | Blood | Chemistry |
| 113 | Glucose | Blood | Chemistry |
| 53 | Alanine Aminotransferase (ALT); Asparate Aminotransferase (AST) | Blood | Chemistry |
| 150 | Phosphate | Blood | Chemistry |
| 95 | Creatinine; Urea Nitrogen | Blood | Chemistry |
| 54 | Albumin | Blood | Chemistry |

Table 7: Test log-likelihood on MIMIC datasets.

| Data | HP-GS | HP-GS-A | HP-LS | NSMMPP | CGPRPP-1 | CGPRPP-10 | CGPRPP* |
|---|---|---|---|---|---|---|---|
| 355 | -3668 | -3947 | -6510 | -3664 | **-3249** | -3374 | **-3249** |
| 60 | -4673 | -5051 | -7299 | -4660 | -4246 | **-4203** | -4246 |
| 3 | **-3721** | -3733 | -5722 | -3737 | -3759 | -3847 | -3759 |
| 95 | -4064 | -4390 | -5712 | -3982 | **-3817** | -3933 | -3933 |
| 368 | -3366 | -3711 | -5625 | **-3309** | -3378 | -3538 | -3378 |
| 354 | -4344 | -4792 | -7185 | -4409 | -4225 | **-3984** | -4225 |
| 151 | -3338 | -3574 | -5323 | -3763 | **-3093** | -3313 | **-3093** |
| 550 | -1053 | -1064 | -1744 | **-1039** | -1175 | -1063 | -1175 |
| 113 | -4656 | -5049 | -7143 | -4539 | -4276 | **-4142** | -4276 |
| 140 | -3206 | -3475 | -4625 | -3244 | -2942 | **-2933** | -2942 |
| 294 | -1011 | -1054 | -1308 | **-941.2** | -993.6 | -1131 | -993.6 |
| 17 | -3783 | -3807 | -5339 | **-3758** | -3808 | -4120 | -3808 |
| 150 | -3238 | -3537 | -4894 | -3377 | **-3100** | -3144 | **-3100** |
| 80 | **-3388** | -3772 | -5365 | -3903 | -3402 | -3426 | -3402 |
| 394 | -3098 | -3251 | -4945 | -3268 | **-3010** | -3127 | **-3010** |
| 1 | **-3220** | -3291 | -3772 | -3228 | -3234 | -3737 | -3234 |
| 53 | -1913 | -2138 | -2963 | -1916 | -1900 | **-1803** | -1900 |
| 7 | **-2502** | -2533 | -3514 | -2626 | -2512 | -2729 | -2512 |
| 8 | **-1633** | -1667 | -3142 | -1786 | -1694 | -1652 | -1694 |
| 18 | -1596 | -1678 | -3085 | **-1532** | -1648 | -1817 | -1648 |

## K  Time prediction evaluation

We also evaluate the performance of our method for predicting the time of each target event on the MIMIC datasets. On each dataset, we repeat the experiment for each method 5 times and show the average results. The setting of each method is the same as for likelihood evaluation, except for HP-LS, where we only test for $h = 2$. We sample 100 times to estimate the expected time to each next event for all the methods.

We evaluate the accuracy of the time predications using root mean square error (RMSE), where the difference between the predicted time and the true time of each event is calculated. The results are in Table 8. The unit is hour. CGPRPP* has the best or close to best results in most cases, except for lab class 8. In that case, CGPRPP ($Q = 10$) is selected over CGPRPP ($Q = 1$) based on the training likelihood, although the latter has a much better time prediction accuracy on the test data.

Table 8: RMSE (hour) of time predictions on MIMIC datasets.

| Data | HP-GS | HP-GS-A | HP-LS | NSMMPP | CGPRPP-1 | CGPRPP-10 | CGPRPP* |
|---|---|---|---|---|---|---|---|
| 355 | 16.43 | 14.78 | 16.98 | 22.06 | **13.79** | 16.59 | **13.79** |
| 60 | 11.81 | 10.4 | 11.19 | 13.2 | **9.956** | 10.6 | **9.956** |
| 3 | 13.46 | **13.45** | 24.75 | 60.44 | 19.13 | 18.85 | 19.13 |
| 95 | 17.43 | 17.17 | 17.36 | **15.99** | 16.17 | 17.43 | 17.43 |
| 368 | 25.32 | 19.24 | 31.7 | 19.71 | **17.03** | 22.18 | **17.03** |
| 354 | 49.1 | 50.13 | 50.46 | **46.87** | 49.61 | 48.52 | 48.52 |
| 151 | 23.36 | 23.3 | 23.53 | 24.12 | **22.37** | 25.91 | **22.37** |
| 550 | 101.4 | 96.11 | 139.3 | 177 | 96.97 | **91.01** | 96.97 |
| 113 | 13.32 | 11.65 | 13.4 | **9.457** | 11.22 | 10.52 | 11.22 |
| 140 | 17.45 | 11.86 | 11.8 | **9.633** | 9.892 | 11.49 | 9.892 |
| 294 | 98.16 | 87.78 | 104.3 | 185.6 | **74.12** | 76.23 | **74.12** |
| 17 | 28.19 | **28.18** | 31.96 | 29.16 | 28.63 | 95.36 | 28.63 |
| 150 | 22.65 | 15.76 | **15.45** | 16.27 | 15.87 | 15.86 | 15.86 |
| 80 | 44.9 | 44.53 | 46.39 | **43.67** | 44.15 | 44.18 | 44.15 |
| 394 | 29.76 | 26 | 46.24 | 30.05 | 25.45 | **21.04** | 25.45 |
| 1 | 25.85 | **25.84** | 32.43 | 26.53 | 25.97 | 29.36 | 25.97 |
| 53 | 51.7 | 36.41 | 59.48 | 38.55 | 27.84 | **27.49** | **27.49** |
| 7 | 24.02 | **22.92** | 34 | 56.59 | 25.71 | 24.54 | 25.71 |
| 8 | 25 | **24.64** | 63.39 | 26.08 | 28.75 | 65.34 | 65.34 |
| 18 | 58.99 | 57.89 | 86.18 | 1611 | 62.2 | **56.49** | 62.2 |

## L    Related work and discussion

### L.1    Hawkes process variants

Hawkes processes [Hawkes, 1971] are one of the most widely used models for event sequences. It models the dependencies between events through so-called triggering kernels. More recently different models have been proposed to generalize Hawkes processes. Zhou et al. [2013] propose to learn the triggering kernels of a Hawkes process nonparametrically after discretization by solving ordinary differential equations. Eichler et al. [2017] also use discretization of the triggering kernels, but their model learns the kernels by solving least-square problems. Xu et al. [2016] propose to use a set of basis functions to approximate the triggering kernel nonparametrically, which does not require discretization, and their method shows better performance than the previous works. Lee et al. [2016] use a stochastic process to model the evolution of the excitations (weights) in a Hawkes process, so the weights become random variables instead of constant parameters. Restricting the kernels to the same exponential kernel (sharing one parameter) for efficiency, they propose a simulation algorithm and an inference algorithm based on a hybrid of MCMC algorithms. Wang et al. [2016] define an isotonic Hawkes process, where the conditional intensity function of the Hawkes process is transformed through a *monotonic* discretized nonparametric link function, and the triggering kernel is given and needs to be continuous and *monotonically decreasing*. The link function is estimated by moment matching instead of traditional maximum likelihood estimation. Zhang et al. [2018] develop a method to nonparametrically estimate the triggering kernel of a *univariate* Hawkes process. Bacry and Muzy [2014] prove a connection between the triggering kernel of a multivariate Hawkes process and its second-order statistics, from which they develop an estimation method of the triggering kernel by solving Wiener-Hopf systems through discretization. Donnet et al. [2018] study nonparametric Bayesian estimation of the triggering kernel of a multivariate Hawkes process, focusing on the theoretical analyses of the posterior convergence rates.

All the above works are only concerned with nonparametric or flexible estimation of a *part* of the conditional intensity function, mostly the triggering kernel. In contrast, our method nonparametrically estimates the *whole* conditional intensity function, which is a major difference that makes the model more flexible. Moreover, our method uses GPs as the nonparametric model for the dependency of the intensity on the history and does not require any discretization. Despite being flexible, the model can be learned efficiently using approximate inference without the need of sampling.

## L.2 GP-modulated point processes

Gaussian processes (GPs) are probabilistic models for functions [Rasmussen and Williams, 2006]. They have been used to model the intensity functions of point processes. These models are also referred to as GP-modulated Poisson processes or GP-modulated Cox processes. Adams et al. [2009] propose an MCMC algorithm for a Sigmoidal Gaussian Cox process (SGCP), where the GP is mapped through a logistic function to the intensity function. Rao and Teh [2011] extend SGCP by augmenting the logistic mapping from the GP to the intensity function with a time dependent function. Lasko [2014] proposes to use the exponential transformation of a GP as the intensity function, so there is no upper bound on the intensity function, and it can model bursty events better than SGCP. In contrast to the previous works, where sampling algorithms are used for inference, Lloyd et al. [2015] propose a variational inference algorithm for GP-modulated point processes, assuming the intensity function is a *square* transformation of a GP. This enables closed-form evaluation of the integral of the intensity function, which is not possible in previous works. Kim [2018] combines Markov modulated Poisson processes with GP-modulated point processes such that the intensity function is a switching model of multiple GPs controlled by a Markov process.

The aforementioned GP-modulated point processes are univariate models, where each sequence consists of points of one type. On the other hand, there are multivariate point processes, where points can have different types. Gunter et al. [2014] extend SGCP to multivariate point processes, where each intensity function is defined as a convolution of a shared set of GPs. An MCMC algorithm is developed for inference. Lloyd et al. [2016] extend [Lloyd et al., 2015] to multivariate point processes, and call their model Latent Point Process Allocation (LPPA). In LPPA, the intensity function of each variate is a positive weighted sum of squared GPs. Ding et al. [2018] further extend LPPA through Dirichlet processes. The intensity function of each variate becomes an *infinite* weighted sum of squared GPs. This resolves the problem of having to pick an appropriate number of latent GPs beforehand in LPPA.

Our method also utilizes the flexibility of GPs. Different from the above works, we do not assume a sequence-specific latent state. Instead our model can be viewed as defining a local latent state relative to the recent events and independent of the sequence. In this way, our model can be applied across sequences, i.e., trained on a set of sequences and make predictions on other unseen sequences, while the GP-modulated point processes cannot.

## L.3 Alternative point-process models

Neural networks have also been studied as a way to model the intensity functions of point processes. Du et al. [2016] develop an RNN for modeling event time series. Event labels and inter-event times are used as the input at each step. Mei and Eisner [2017] draw intuition from Hawkes processes and develop a continuous-time LSTM, where the memory cell has an exponential decay between two consecutive events. Monte-Carlo sampling is used for evaluating the integral of the intensity function in both training and prediction. Xiao et al. [2017] define a Wasserstein distance for point processes and combine it with Generative Adversarial Networks (GANs) to train generative models. Their work is limited to univariate point processes. Li et al. [2018] propose a reinforcement learning framework for learning generative point processes models. Their work is also limited to univariate point processes. In contrast to neural-network-based methods, our approach uses nonparametric Bayes to provide flexibility. It is a more principled approach, and similar to Hawkes processes, the models are easier to explain and understand.

Another type of point-process models are based on featurization of the history. Gunawardana et al. [2011] assume the intensity function is a piecewise-constant function of the past events, which is a decision tree mapping features extracted with window-based functions from the past events to constants. Lian et al. [2015] also assume a piecewise-constant intensity function but extend it to multitask problems using a hierarchical model. Different from these works, our model learns the dependencies between events directly from the data and does not require any feature engineering.

## Footnotes

[1] `https://github.com/HongtengXu/Hawkes-Process-Toolkit`