[Reviews · NeurIPS 2019]

Reviewer 1



The paper considers how to create a flexible method for modelling Hawkes-like processes with flexibility in the triggering kernel, using Gaussian processes working on a different input than is usually attempted. Section 1-3 is well written, very clear and gives a good motivation and description of how to get a GP regressive point process. The idea in the preliminary work that the likelihood of a Hawkes process factorises across different types is surprising as the triggering kernel in the CIF appears to connect $\lambda_{u_{i}}(t)$ and $\lambda_{u_{j}}(t)$, since $u_{i}$ and $u_{j}$ appear to be part of a single triggering kernel, and hence are connected. Can the authors clarify, is there some conditional independence I am not spotting here? It doesn't appear to be of significance however as the rest of the paper only focusses on a single event type in the end. The idea of introducing an additional kernel working on some augmented data in the form of the evaluation of an indicator function is a clever one that skirts the issue that certain contributions would otherwise be undefined. I believe such a kernel could be used in other applications and as such is, as far as I am aware, a novel and useful contribution. Section 4, the most significant methodological section, lacks clear motivation however. It isn't clearly discussed why a conditional distribution approach is needed in the first place. Why do we need such a conditional GP model instead of using the kernel defined in section 3 and following Lloyds derivation from there? It is indicated that this is to 'encode the dependencies of the CIF on the past events', but I think this could really do with clarifying. Perhaps the authors could expand on this in both their rebuttal and in any revision of the manuscript? In Section 4, it appears odd to introduce additional noise into f_{Z} without justification, rather than simply using equation (4). I presume this is because it is intractable in (4) but the additional noise results in it being tractable in (6). However it is not immediately clear why this is done, or what the resulting implication are. For example the draws from f_{Z} are now 'rough' rather than smooth, and not differentiable, this doesn't seem like desirable behaviour. Is S_{\epsilon} learnt to be very small or is it reasonably large? Presumably this would effect the quality of the approximation. Can the author clarify why this is done and illustrate the resulting implications on the approximation being made? The synthetic experiments illustrate that the triggering kernel can be learnt well non-parametrically and a brief discussion illustrate that it is able to model triggering kernels that contain both excitation and inhibition, a property most Hawkes process approximations don't have. I appreciate the authors efforts of doing a relatively in-depth study of the models properties here rather than just providing results on a bunch of experiments. For the IPTV it's not clear why CGPRPP should not be able to model bursty events. If it's non-parametric shouldn't it adapt to this situation given enough inducing points? Given CGPRPP's apparent flexibility but (slightly) worse performance, it feels like some additional digging for some reasons would be useful if the answer isn't clear. Is the model overfitting? Does it need additional kernel contributions for flexibility or periodic components? Do the inducing inputs need to be placed more strategically? Are simply more of them needed? For the MIMIC data there is a nice illustration for 331 showing inhibition that GP-GS can't model, but there is no explanation of what goes *wrong* in the classes the CGPRPP seems to struggle with. I think this would be enlightening. It is ok to not always give ground-breaking improvements in predictive performance, but it is useful to know *why*. I think what is sorely missing however is a comparison to at least one method that is more similar, and not parametric, to show that the method is competitive. For example [Rousseau et al], [Xie et al.], etc. It seems unfair to only compare to relatively inflexible models when more flexible ones already exist. Minor comments: 167. It should be made clear in the main text what 'some restrictions' are. 238. Says HP-GS and CGPRPP perform the same, but that doesn't seem quite true, in fact CGPRPP marginally outperforms HP-GS here. [Efficient Non-parametric Bayesian Hawkes Processes. R Zhang, C Walder, MA Rizoiu, L Xie.] [Nonparametric Bayesian estimation of multivariate Hawkes processes. S Donnet, V Rivoirard, J Rousseau]

Reviewer 2



Originality: the paper presents a novel approach of combining GPs and point processes. The previous work is adequately cited. Quality: the submission seems sound, but I would like to have a clarification about the new introduced kernel. Clarity: the paper is clear and well written, but clarification is necessary about Eq.3. Significance: the paper presents some important theoretical contributions

Reviewer 3



The authors present a nonparametric model for regressive PP based on GPs, where the GPs are used as a prior model for the intensities of the PP. My research field is GPs and I am not familiar on PP to fully assess the novelty of this work. However, to the best of my knowledge this work is indeed novel, it connects with sufficient material in the literature to justify itself and provides convincing theoretical analysis and experimental validation. It is a novel, sound and well written work. I have a couple of suggestions to further improve this work detailed in the next answers

Reviewer 4



Originality: The work proposed in the paper is original in a way that they propose a GP for modeling PP where the kernel of GP is a function of inter-arrival times instead of the absolute time of the event in the sequence. They further develop a methodology to learn the GP based on latent points in an efficient manner. Clarity: The paper is well written and easy to follow mostly apart from some discrepancies as suggested below in improvements. Most of the differences from existing work and novelty in the paper are well laid out. Still paper puts a huge burden on the reader to go through a massive amount of supplementary material to fully understand the working of the learning procedure, it is the main contribution and at least be intuitively explained how is it derived. The experiments section clearly explains the parameters used for competitive methods and the proposed methods to the extent that the reader can infer them after looking them in the code provided.

[Author Response · NeurIPS 2019]

We would like to thank all the reviewers for the insightful and detailed comments.

**Reviewer 1**  Each triggering kernel $\phi_{u_i u_j}$ in HP only connects *past* events of type $u_j$ to the CIF, $\lambda_{u_i}(t)$, of type $u_i$. The CIF by definition is *conditioned* on the history (all types of events that occur before $t$), so $\lambda_{u_i}(t)$ only depends on the history, not on $\lambda_{u_j}(t)$. The factorization can be derived directly from the general form of the probability density (line 68-71) due to the definition of CIF (line 63) and is not restricted to HPs. Also, we derive the method for one **target** event type, but model the dependencies between that type and *all* types of events through $x(t)$ (line 80-91): we model each $\lambda_u(t), u = 1, \ldots, U$, with a GP, but the input $x(t)$ of the GP depends on all $U$ types of events that occur before $t$.

A key difference between conditional GP and variational sparse GP [Titsias 2009, Lloyd et al. 2015] is in the flexibility of the models. Inducing points in variational sparse GP are marginalized out in the final inference, so they do not add any flexibility to the model (they only improve computational efficiency). Conditional points in conditional GP are kept in the final inference as being conditioned on (e.g., Eq. 4), so they add flexibility and therefore help the model to store the dependency information learned from the data. We have results comparing conditional GP and variational sparse GP in the supplementary material (Section F).

The dependency between events can have variance in the data: given the same history, the CIF can still vary in different realizations. Noiseless conditional points cannot capture the variance. That is why we introduce noise in $f_Z$, generalizing the noiseless version. $S_\epsilon$ learned from the data captures the variance and covariance of the conditional points. The entries in $S_\epsilon$ are small in many cases, but it is nice to allow it to adapt to the data (e.g., to have large entries when the dependency between events shows much variance in the data).

We apologize for the ambiguity. We did not mean CGPRPP cannot model bursty events, but tried to stress that it is easier for HP variants to model them, since the data satisfy their assumption (bias). The majority of the inferior results on IPTV and MIMIC are seemingly caused by overfitting (CGPRPP has better training likelihood), the root cause of which could be change in data distribution between training and test sets.

We note that both HP-GS and HP-LS are *nonparametric* and flexible, which is why we picked them as baselines. Thanks for pointing out other works. However, 1) [Xie et al.] only consider *univariate* event sequences, so their method cannot be applied to *multivariate* event sequences we consider. 2) [Rousseau et al.] focus on theoretical analyses of posterior convergence rates. Although they have a "numerical illustration", they do not evaluate predictive performance nor compare against existing methods. The inference algorithm is only briefly mentioned lacking details. 3) Similar to HP-GS [Xu et al.], in both [Xie et al.] and [Rousseau et al.], only the *triggering kernels* are nonparametric, while the CIF has the same form as HPs, so they also cannot model a mix of excitations and inhibitions. In contrast, CGPRPP can model it, because the CIF is nonparametric, so CGPRPP has the same advantage. 4) None of them compare against HP-GS or HP-LS in the experiments, so their methods are not necessarily more SOTA than HP-GS or HP-LS.

**Reviewer 2**  We believe you mean Eq. 1 that defines the kernel. As noticed, $x_d(t) = t - s_u^q(t)$ is undefined when the $q$-th (from last) event of type $u$ does not exist yet at time $t$ (e.g. when there is no type $u$ event in the history). Conceptually, we augment each dimension $x_d(t)$ with a new dimension $x_{D+d}(t)$: $x_{D+d}(t) = 0$ if $x_d(t)$ is undefined and 1 otherwise, increasing the dimensionality of $x(t)$ from $D$ to $2D$. Then instead of leaving $x_d(t)$ undefined, we can assign a special value $x_d(t) = \infty$ (or a very large number), when the $q$-th (from last) event of type $u$ does not exist. Then define $\mathbb{I}[x_d(t)] = 0$ if $x_d(t) = \infty$ and 1 otherwise, or equivalently define $\mathbb{I}[x_d(t)] = x_{D+d}(t)$. The overall kernel is valid due to the kernel composition rules and that $K_1$ and $K_2$ are valid kernels on augmented dimensions and original dimensions. $K_2$ is widely used. $K_1$ is a valid kernel because it is an inner product on augmented dimensions.

**Reviewer 3**  We cut the conclusion text due to the space limit. We plan to work on the text and hope to find the space to add it back and also add an illustration. We will fix the inconsistency in the bibliography (thanks for noticing it).

**Reviewer 4**  By "focus on one type" we mean deriving the method for one **target** event type, but the dependencies between that type and *all* types of events are modeled through $x(t)$ (line 80-91): we model each $\lambda_u(t), u = 1, \ldots, U$, with a GP, but the input $x(t)$ of the GP depends on all $U$ types of events before $t$. Based on the factorization of the density (line 68-71), we can repeat the derivation for all target event types (same equations with different target event types). Losses (negative log-likelihood) from different sequences are summed for learning (Eq. in line 79 with $\log$).

In Remark 1, we mean it is isolated from the time *denoted by* $t$ as in $x(t)$, which is the *absolute* time. We will make it clearer in the paper. We note that whether Assumption 1 breaks the mechanics of HP depends on what triggering kernel is used for HP. For example, if the triggering kernel is only nonzero within a bounded interval and zero otherwise, i.e., $\phi(t) = 0$ if $t > A, A \in \mathbb{R}_{>0}$ (interestingly [Donnet et al.] you referenced assume this (Section 1.4) and state it is a "very common" hypothesis (Section 2.1)), then Assumption 1 will not break the mechanics. Thanks for pointing out other works. We will cover them in related work.

[Meta-Review · NeurIPS 2019]

The reviewers had low confidence in their assessments. Overall, I think that the idea as presented is sound and novel. However, the clarity of presentation could be significantly improved: reviewers found key parts of the exposition to be lacking in motivation; Eq 3 was not clear to one reviewer; and another reviewer felt that the details of the learning procedure should have been included in more detail in the paper rather than in the supplementals. I also think the extension to models for multiple types needs to be made more explicit.